# The Activation of Prothrombin Seems to Play an Earlier Role than the Complement System in the Progression of Colorectal Cancer: A Mass Spectrometry Evaluation

**DOI:** 10.3390/diagnostics10121077

**Published:** 2020-12-11

**Authors:** Maider Beitia, Paolo Romano, Gorka Larrinaga, Jon Danel Solano-Iturri, Annalisa Salis, Gianluca Damonte, Marco Bruzzone, Marcello Ceppi, Aldo Profumo

**Affiliations:** 1Department of Physiology, Faculty of Medicine and Nursing, University of the Basque Country (UPV/EHU), 48940 Leioa, Biscay, Spain; maider.beitia@ucatrauma.com (M.B.); gorka.larrinaga@ehu.eus (G.L.); 2Proteomics and Mass Spectrometry Unit, IRCCS Ospedale Policlinico San Martino, 16132 Genova, Italy; aldo.profumo@hsanmartino.it; 3Department of Nursing, Faculty of Medicine and Nursing, University of the Basque Country (UPV/EHU), 48940 Leioa, Biscay, Spain; 4Cancer Biomarkers Group, BioCruces-Bizkaia Health Research Institute, 48903 Barakaldo, Biscay, Spain; jondanel.solanoiturri@osakidetza.eus; 5Department of Pathology, Donostia University Hospital, 20014 San Sebastian-Donostia, Gipuzkoa, Spain; 6Department of Medical-Surgical Specialities, Faculty of Medicine and Nursing, University of the Basque Country (UPV/EHU), 48940 Leioa, Biscay, Spain; 7Centre of Excellence for Biomedical Research (CEBR), University of Genoa, 16132 Genova, Italy; annalisa.salis@unige.it (A.S.); gianluca.damonte@unige.it (G.D.); 8Department of Experimental Medicine (DIMES), University of Genoa, 16132 Genova, Italy; 9Clinical Epidemiology Unit, IRCCS Ospedale Policlinico San Martino, 16132 Genova, Italy; marco.bruzzone@hsanmartino.it (M.B.); marcello.ceppi@hsanmartino.it (M.C.)

**Keywords:** serum, prothrombin, complement, peptidome, mass spectrometry, colorectal cancer

## Abstract

Colorectal cancer (CRC) is the second cause of death in men and the third in women. This work deals with the study of the low molecular weight protein fraction of sera from patients who underwent surgery for CRC and who were followed for several years thereafter. MALDI-TOF MS was used to identify serum peptidome profiles of healthy controls, non-metastatic CRC patients and metastatic CRC patients. A multiple regression model was applied to signals preliminarily selected by SAM analysis to take into account the age and gender differences between the groups. We found that, while a signal m/z 2021.08, corresponding to the C3f fragment of the complement system, appears significantly increased only in serum from metastatic CRC patients, a m/z 1561.72 signal, identified as a prothrombin fragment, has a significantly increased abundance in serum from non-metastatic patients as well. The findings were also validated by a bootstrap resampling procedure. The present results provide the basis for further studies on large cohorts of patients in order to confirm C3f and prothrombin as potential serum biomarkers. Thus, new and non-invasive tests might be developed to improve the classification of colorectal cancer.

## 1. Introduction

Colorectal cancer (CRC) is one of the most common cancers in the world. Even if its death rate has dropped due to the screening techniques, it is still the second cause of death in men and the third in women [1]. Early detection of CRC remains a challenge and the identification of new molecular signatures represents an important field of translational research. Indeed, patients who receive an early diagnosis of CRC and quickly undergo surgery have a favourable five-year survival rate. However, many CRC cases are diagnosed at a late stage with dramatic consequences on both therapy and survival rates [2,3].

In recent years, several authors have described the study of the low molecular weight fraction of human serum for diagnostic and/or prognostic purposes in several diseases, including cancer [4,5,6]. In fact, it has been proposed that the low molecular weight fraction of the serum proteome may be correlated with physio-pathological events occurring in all districts of the organism [7,8]. The possibility of evaluating changes in the serum composition associated to CRC onset and progression could be imagined as a valid alternative to the invasive colorectal biopsy for the future.

Mass spectrometry (MS) is well suited for the detection of proteins or peptides that might not be detectable with traditional techniques and that might represent a dynamic reflection of tissue function. In 2012, Fan and co-workers identified a peptide signature containing two peaks, at m/z 741 and 7772, able to distinguish the CRC patients from healthy controls [9]. Recently, Wang et al. described a study on human serum peptidome for diagnostic purposes in colorectal cancer [10]. The authors proposed a diagnostic panel based on five peptides able to discriminate CRC patients from healthy controls (peaks at m/z 1895.3, 2020.9, 2080.7, 2656.8 and 3238.5). Three peaks, having m/z 1895.3, 2020.9 and 3238.5 were identified as the partial sequences of complement component 4 (C4), complement component 3 (C3) and fibrinogen alpha chain (FGA), respectively. However, to the best of our knowledge, there are no studies describing a peptidomic signature able to distinguish between different stages of the disease.

In this study, we used SeraDeg [11] to select the serum samples that were sufficiently preserved to be included into the experimental groups. After the selection, a comparative analysis has been performed by significant analysis of microarrays (SAM) [12] in order to identify a peptidomic signature able to distinguish between healthy controls, non-metastatic CRC patients and metastatic CRC patients.

## 2. Materials and Methods

Acetonitrile, methanol and water were LiChrosolv solvents obtained from Merck (Darmstadt, Germany). Trifluoroacetic acid (TFA) and alpha-cyano-4-hydroxycinnamic acid (CHCA) were obtained from Fluka (Sigma-Aldrich, St. Louis, MO, USA).

### 2.1. Patient Selection and Ethical Aspects

Blood samples from 180 patients with CRC and 70 healthy volunteers were obtained. Samples were collected between 2012 and 2016, prior to surgery, and aliquots were stored at −80 °C until processing. Moreover, as the duration of cryopreservation could potentially affect the protein profile, as it was previously evidenced by us [13], cases and controls were also rigorously matched for the length of cryopreservation period. The CRC serum samples from patients were prepared from blood specimens collected before the start of any anticancer treatment. The samples were collected in the Basque Biobank for Research following a strict and standardized protocol to ensure the appropriate processing and storage that preserves a correct protein profile. Samples were obtained from peripheral blood. Tubes were centrifuged at 2500 rpm for 20 min at room temperature and placed in 500 μL aliquots.

The assessment of metastatic and non-metastatic patients was based on the “Protocol for the Examination of Resection Specimens From Patients With Primary Carcinoma of the Colon and Rectum” of the College of American Pathologists (2020) [14].

All the experiments carried out in this study comply with the current Spanish and European Union legal regulations. Samples and data from patients included in this study were provided by the Basque Biobank for Research—OEHUN (http://www.biobancovasco.org/). All patients at Basurto University Hospital were informed about the potential use for research of their biological samples and accepted this eventuality by signing a specific document approved by the Ethical and Scientific Committees of the Basque Country Public Health System (Osakidetza) (CEIC 11-51).

### 2.2. Solid Phase Extraction

Functionalised super-paramagnetic beads (DynabeadsVR RPC 18 Life Technologies Dynal, Irvine, CA, USA) with a C18 alkyl-modified surface were used to prepare a low molecular weight (LMW) protein fraction from serum. Serum sample preparation was performed as described in detail in a previous paper [15]. Some modifications were applied to the DynabeadsVR RPC 18 manufacturer’s protocol. Briefly, 40 μL of the bead suspension were washed once with water and thrice with 100 μL of 200 mM NaCl, 0.1% TFA. The beads were re-suspended in 20 μL of water, mixed with 50 μL of serum and incubated at room temperature for 5 min. After incubation, the tube was placed in a magnetic concentrator (Dynal MPCVR, Invitrogen Dynal, Oslo, Norway), and the supernatant was discarded. The bead-peptide complex was washed thrice with 300 μL of 0.1% TFA in water, and the bound peptides were eluted by incubation at room temperature for 2 min with 12 μL of a 1:1 acetonitrile/water solution containing 3.5 pmol/μL of a synthetic internal standard peptide (MW 1419.76). This standard peptide, which was later used to normalize the data, was directly spiked in the eluting solution in order to avoid any interference during the binding reaction between analytes and the magnetic beads.

### 2.3. API-MALDI/TOF MS Analysis

The target plates for the API-MALDI/TOF analysis were prepared following the standard procedure previously cited [4]. Briefly, one volume of the eluted sample was mixed with two volumes of premade matrix solution (6.2 mg of CHCA in 1 mL of 36% methanol, 56% acetonitrile and 8% water). Next, 1 μL of this mixture was spotted in quadruplicate onto the MALDI target plate and allowed to dry at room temperature in a vacuum desiccator. API-MALDI/TOF analysis was performed in positive mode on a 6210 Time of Flight LC/MS (Agilent, Santa Clara, CA, USA) coupled with an atmospheric pressure PDF-MALDI Ion Source (Agilent) equipped with a 337 nm nitrogen laser. The following voltages were applied: fragmentor 300 V, skimmer 60 V, OCT RF 300 V. The acquisition laser power was set at 35% of maximum (peak power 75 kW, pulse energy 300 mJ). Data acquisition was automated using Mass Hunter software (Agilent) and programmed to accumulate 600 shots per spectrum. The irradiation program was automated using the spiral motion control of the PDF-MALDI ion source. The instrument was externally calibrated using Tuning Mix (Agilent). The nominal resolution of the instrument was 20.000 (17.000 observed), and the nominal mass accuracy (with internal calibration) was <2 ppm.

### 2.4. Nano-LC Mass Spectrometry

An aliquot of the DynabeadsVR RPC 18 extracted peptides was analysed by nano-HPLC-MS/MS using an Ultimate 3000 nano-HPLC system managed by the CHROMELEON Chromatography Data System (CDS) software (Thermo Scientific, Oslo, Norway) connected to a Hybrid Quadrupole-Orbitrap mass spectrometer (Q Exactive, Thermo Scientific).

The peptides were re-suspended immediately before analysis and were first loaded onto a trapping column (Acclaim PepMap C18, length 20 mm, diameter 0.1 mm, particle size 5 μm, pore size 100 Å, Thermo Fisher Scientific, Oslo, Norway) using the loading solvent (5–95% acetonitrile/water + 0.05% trifluoroacetic acid) at a flow rate of 5 μL/min for 5 min. The trapping column was then switched in-line with the separation column, and the peptides were eluted with increasing organic solvent at a flow rate of 300 nL/min. Mobile phase A was 0.1% formic acid in water; mobile phase B was 95–5% acetonitrile/water + 0.08% formic acid. The separations were carried out at 35 °C using an Easy spray column (PepMap RSLC C18, length 150 mm, diameter 0.075 mm, particle size 3 µm, pore size 100 Å, Thermo Scientific) applying a linear gradient from 4% to 95% of eluent B in 55 min.

All the analyses were carried out in the positive ion mode in a mass range between 395 and 2000 m/z. Single MS survey scans were performed in the Orbitrap using a maximal ion injection time of 100 ms. The resolution was set to 70,000, and the automatic gain control was set to 3 × 106 ions. The experiments were performed in data-dependent acquisition mode with alternating MS and MS/MS experiments. The minimum MS signal to obtain MS/MS was set to 500 ions, with the most prominent ion signal selected for MS/MS using an isolation window of 2 Da. The m/z values of signals already selected for MS/MS were put on an exclusion list for 5 s using dynamic exclusion. In all cases, one micro-scan was recorded. Collision-induced dissociation (CID) was done with a target value of 5000 ions, a maximal ion injection time of 50 ms, normalized collision energy of 35%. A maximum of 10 MS/MS experiments/MS scan were performed.

Raw MS files were processed with the Thermo Scientific Proteome Discoverer software version 1.4. Peak list files were obtained by the SEQUEST search engine against the Homo Sapiens database containing both forward and reversed protein sequences. The resulting peptide hits were filtered for a maximum 1% FDR (false discovery rate) using the percolator tool. The peptide mass deviation was set to 10 ppm, and a minimum of six amino acids to identify peptides were required. The database search parameters were: mass tolerance precursor 20 ppm, mass tolerance fragment CID 0.8 Da with dynamic modification of deamidation (N, Q), oxidation (M). For all searches, the option trypsin with two missed cleavages was selected.

### 2.5. API-MALDI/TOF MS Data Format Conversion

Raw data generated by the Qualitative Analysis software (Agilent Technologies, Santa Clara, CA, USA) were converted from the proprietary format (.d) to the mzML standard format by using MSconvert, a tool of the software suite ProteoWizard [16]. Four spectra, corresponding to the quadruplicated spots, were then determined for each sample by averaging 15 consecutive scans from the original data by means of the MALDIquant R package [17].

### 2.6. Sample Selection

All serum samples were tested for their integrity by means of SeraDeg [11], a publicly available web tool aimed to assess the quality of sera through a comparative analysis of the contents, in the related MS spectra, of fibrinopeptide A (fpA) fragments, a molecule which is known to be easily degraded under physico-chemical stress conditions. For each sample, SeraDeg provides quality scores (SeraDeg quality score—SDQS) that allow researchers to assess the quality of samples and discard those samples that do not have an adequate quality.

This score is mainly based on the overall abundance of fpA fragments, which is compared to the overall abundance of the same molecules in a reference spectrum derived from a sample of assessed quality and the ratio between the percentages of high-mass (m/z 1350.6 and 1465.63) and low-mass (m/z 905.46 and 1077.53) fpA fragments.

The spectra created during the previous step by using MALDIquant were exported in text format (two columns: one for the m/z value, one for the relative abundance) and included in a compressed zip archive in order to analyse them with SeraDeg. The following thresholds were applied for the selection: overall abundance of fpA fragments greater than 33.33% of the overall abundance in the reference spectrum, percentage of high-mass fpA fragments greater than 66.67% of the percentage of low-mass fpA fragments.

According to the results of this analysis, only a subset of the samples that were initially acquired was finally used in the following statistical analysis.

### 2.7. Spectra Pre-Processing

Lists of peaks were determined for each spectrum by applying a two-step procedure. All peaks were first determined by selecting local maxima (i.e., values in the given spectrum higher than both the previous and the following values). The final list of peaks was then determined by applying a median filter (half window size of 20 and signal-to-noise ratio of 2.5).

The lists of peaks were then analysed by means of Geena 2, a publicly available web tool for pre-processing MALDI/TOF spectra that, among other functions, is able to:Normalize both abundances and adjust m/z values against a reference signal;Sum up isotopic peaks of the same molecule;Average technical replicas from each sample, thus producing a single spectrum that is representative for the sample;Align representative spectra and produce a table of aligned signals with corresponding abundances.

Both the average and alignment steps above allow discarding those signals that are included only in a few spectra, so that sporadic signals can be removed from the output. For the average step, we adopted a high threshold (at least three spectra out of the four replicas must include the signal for its inclusion in the alignment). For the alignment step, instead, we had to define a low threshold in order to prevent that signals present only in one of the groups under analysis were discarded. The smaller group led this choice. We set this threshold to five: only signals found in at least five representative spectra were included in the final alignment.

### 2.8. Statistical Analysis

The output generated by Geena 2 was then analysed by means of SAMR, in order to identify signals significantly different in the groups, i.e., healthy volunteers, patients affected by metastatic CRC and patients affected by non-metastatic CRC. The multiple regression model [18] was applied to peptides eventually selected by the SAM analysis to take into account the age and gender differences between the two groups of patients and controls. The parameters estimated by this model can be interpreted as the difference in the mean values of a peptide between patients and controls. Given the small size of our sample, we chose to obtain a robust estimate of the standard errors of the model parameters by using a non-parametric bootstrap.

### 2.9. Data Validation

A bootstrap resampling procedure was implemented in order to validate the output of the SAM analysis. The bootstrap method consists in the construction of new datasets by randomly resampling from the original one [19]. Bootstrap may be implemented with or without replacement. In the former case, the new datasets are created by selecting samples from the original data set without any supervision and may, therefore, include any of the original samples more times. Additionally, with replacement of the size of the new datasets may be equal or even greater than the size of the original dataset, thus including more samples than originally. In the latter case, instead, any replication is avoided, and the size of new datasets must strictly be lower than the original size.

Due to the relatively small number of samples involved in our analysis, we preferred to avoid reducing the number of samples in the validation datasets, and we, therefore, applied bootstrap with replacement. Moreover, we choose to keep the size of the new datasets identical to the size of the original dataset, thus avoiding an excessively high number of duplicated samples in the datasets.

By bootstrapping with replacement from the original dataset, we thus created 100 new datasets of the same size of the original one and executed the SAM analysis for each of them. The original ratio between group sizes was also conserved. Finally, we evaluated the frequency of occurrence of the peptides selected as significant by SAM on the original dataset in the results of the new 100 SAM analyses. 

## 3. Results

In this study, we initially analysed the peptidomic profile of 250 serum samples (95 metastatic CRC patients, 85 non-metastatic CRC patients and 70 healthy controls).

### 3.1. API-MALDI/TOF MS Analysis

To determine the low-molecular-weight (LMW) protein profile spectra, the whole sample set was analysed by API-MALDI/TOF mass spectrometry performed on the LMW fraction of sera. The good quality of our MS spectra can be appreciated in Figure 1.

In order to avoid procedural biases, samples were randomly distributed during processing. All spectra were acquired in quadruplicate in a mass range from 800 to 3000 m/z.

### 3.2. Selection of Samples

After the acquisition, the API-MALDI/TOF spectra were tested with SeraDeg in order to avoid possible biases due to altered preservation conditions. As previously reported, this operation selected a set of samples of adequate quality and preservation including 172 pathological samples, of which 92 patients had been diagnosed with metastatic CRC, 80 patients had been diagnosed with non-metastatic CRC, and 19 samples were from healthy volunteers (Table 1).

### 3.3. Selection of Candidate Signals

The sample set selected by SeraDeg was used as a discovery cohort to identify possible candidate peptides able to distinguish controls from patients with reference to metastatic spreading. After the pre-processing steps carried out by means of Geena 2 [20], a total of 122 peaks were detected. Among these, 19 peaks were found to be related to fibrinopeptide A or B and were not considered in the following analyses. Indeed, fpA fragments were used, as previously stated, by SeraDeg for the selection of less degraded samples and consequently taking them into account could erroneously drive the final outcome. The number of useful signals was finally reduced to 103. The MS Excel files including alignment data generated by Geena 2 as prepared for the SAM analysis for the comparison between controls and non-metastatic samples and between controls and metastatic samples are included as Appendix A.

The significance of the SAM analysis was assessed by using the *q*-value that is similar to the well-known *p*-value adapted to the situation where a large number of comparisons is carried out, which is our case since we evaluated 103 peptides. The significance threshold for the *q*-value is 5%. When we compared the three groups of samples, the SAM analysis highlighted only three signals (m/z 1561.72, 2021.08 and 2399.17) having a *q*-value of 5% or less either in the comparison between controls and non-metastatic samples or between controls and metastatic samples (see Table 2). No significant signals were found in the comparison between samples from non-metastatic and metastatic patients.

Indeed, a signal at m/z 1051.71 was also highlighted by the SAM analysis as a significant signal (*q*-value < 5%) in both comparisons, with a fold change of 2.5 for controls versus non metastatic samples and of 2.7 for controls versus metastatic samples. However, this signal is known to be derived from a polymer contaminant from either polypropylene or polyethylene plastic tubes [21]. Consequently, it has been excluded from any further consideration. 

Only one of the other signals, the one at m/z 1561.72, was significantly higher in non-metastatic CRC patients when compared with healthy controls (fold change 16.4). When the healthy controls were compared with the metastatic CRC patients, beyond the 1561.72 signal (whose fold change increased to 17.8), the other two signals (at m/z 2021.08 and 2399.17, with fold change 2.9 and 6.9, respectively) were also found significantly increased. These results are also represented in the form of dot plots in Figure 2 and Figure 3.

In Figure 2, the comparison between controls and non-metastatic patient is shown, while the comparison of controls and metastatic patients is shown in Figure 3. For each of the three significant signals, m/z 1561.72, 2021.08 and 2399.17, the abundance values of the signal in controls’ and patients’ spectra are shown separately.

### 3.4. Identification of the Selected Serum Peptides by Nano-HPLC-ESI-MS/MS

In order to identify the origin of the selected candidate signals, a tandem mass spectrometry experiment was performed analysing a CRC serum sample that has one of the highest abundances of these molecules. Following the submission of the data obtained by the LC separation and MS/MS analysis to the SEQUEST search engine [22] against the Uniprot Homo Sapiens database, the amino acid sequences of peptides m/z 1561.72 and m/z 2021.08 were, respectively, determined, with high confidence, as belonging to prothrombin (Uniprot ID: P00734, sequence: “TATSEYQTFFNPR”) (Appendix A) and to the fragment C3f of complement system C3 (Uniprot ID: P01024, sequence: “SSKITHRIHWESASLLR”) (Appendix A), while the signal 2399.17 was identified as a double α-CHCA matrix adduct (with molecular weight 189.04 Da) to the peptide at m/z 2021.08 (C3f). A signal at m/z 2210.13 corresponding to a single α-CHCA matrix adduct to the C3f peptide at m/z 2021.08 was also identified, but it was not considered significant by the SAM analysis.

### 3.5. Validation by Bootstrap Resampling

As previously reported, bootstrap resampling with replacement was adopted in order to validate the SAM analysis. Starting from the original dataset, we first created 100 new datasets of the same size, and we then analysed them by using SAMR, an implementation of the SAM statistical test based on the R language [23]. It is worth stressing that all results were congruent with the SAMR analysis, i.e., signals identified as increased between controls and non-metastatic patients were also found increased at bootstrap, and vice versa. The same occurred when the analysis was applied to signals in the other comparison, i.e., between controls and metastatic patients.

In Table 3, the frequency by which each peptide identified as significant by the SAM analysis on the original dataset was also identified as significant in the new datasets is reported. In this table, the results of the bootstrap analysis are reported separately for the comparison between controls and non-metastatic patients and between controls and metastatic patients. In both cases, the molecular weight of the signal, the *q*-value of the SAMR analysis on the original data set and the number of occurrences the signal has been considered significant (*p* < 0.05) in the repeated SAMR analyses on the new datasets are reported.

As expected, significant signals (*p* < 0.05) in the SAM analysis on the original dataset have a higher number of occurrences. In particular, the signals of the greatest interest (m/z 1561.72 and m/z 2021.084) have high/very high number of bootstrap occurrences, ranging from 70/100 for m/z 1561.72 in the comparison between controls and non-metastatic patients to 95/100 for both signals in the comparison between controls and metastatic patients.

Finally, Table 4 shows the results from the multiple regression models in order to validate the findings of the SAM analysis taking into account the differences in age and gender between patients and controls. Actually, the increases in abundance observed among patients compared to controls for the selected peptides are confirmed.

## 4. Discussion

This study aimed at the construction of a serum peptide signature able to differentiate the three evaluated populations (metastatic CRC patients, non-metastatic CRC patients and healthy volunteers). The results of the SAM analysis on mass spectrometry data put in evidence three signals showing statistically significant differences, which were able to distinguish between healthy volunteers and CRC patients. 

This peptide signature includes the m/z 1561.72 (human pro-thrombin P00734, fragment “TATSEYQTFFNPR”), m/z 2021.08 (C3 complement P01024, C3f complement fragment “SSKITHRIHWESASLLR”) and m/z 2399.17, identified as a double α-CHCA matrix adduct to the peptide at m/z 2021.08 (C3f). Obviously, since the signal m/z 2399.17 has been found to be an instrumental artefact, its biological meaning is absolutely superimposed on that of the signal m/z 2021.08, and it will not be further discussed.

Our results show that the intensity of the two selected peptides was higher in the serum samples from CRC patients than in those from healthy controls. Interestingly, while the difference for m/z 2021.08 was only found significant between healthy volunteers and metastatic CRC, the difference for m/z 1561.72 was found significant between healthy volunteers and non-metastatic CRC too. Moreover, since other authors described a dependence of complement levels on age and gender in the healthy population [24], we confirmed our results for all three peptides by applying a multiple regression model to adjust data by age and gender.

Cancer is a multifactorial disease whose onset is related to both genetic and environmental factors. So, from the physio-pathological point of view, the increase of pro-thrombin fragment at m/z 1561.72 and full length C3f (m/z 2021.08) should be seen in a broader context taking into account many other factors. Notwithstanding this premise, both the coagulation process and activation of the complement system have been related, in the past, to carcinogenesis [25,26]. Pro-thrombin (factor II), which is an essential component of the blood coagulation mechanism, is transformed into thrombin by pro-thrombinase; then, thrombin transforms fibrinogen into fibrin, which, in combination with platelets, forms the clot able to stop the bleeding. However, apart from the canonical task that it plays in blood coagulation, thrombin has been shown to also affect the carcinogenesis process. Adams and co-workers showed that pro-thrombin plays a broad role in the progression of colonic adenocarcinoma that involves both promoting the growth of established tumours and the potential of circulating tumour cells to form distant metastases [27]. Other authors established that the role of thrombin in cancer pathogenesis is not limited to late events, such as metastasis. Thrombin can also strongly influence the development of pre-cancerous lesions and subsequent tumour formation in the context of inflammation-driven colon cancer [28]. With regard to other neoplasms, thrombin is one of the main coagulation-related proteins increased in pancreatic cancer [29,30,31], while a signal of 4204 Da, identified as a fragment of pro-thrombin by MALDI-TOF/TOF mass spectrometry, has been found significantly increased in patients affected by biliary tract cancer [32].

Zain et al. found a concentration-dependent dual effect of thrombin on impaired growth/apoptosis in tumour cells, mediated by the tumour cell PAR-1 thrombin receptor [33]. They demonstrated a clear thrombin-enhanced tumour cell growth at relatively low thrombin concentration, but, at the same time, that prolonged thrombin exposure leads to cell cycle arrest and apoptotic effect. Other studies indicate that high thrombin concentrations are able to induce the caspases 1, 2 and 3, whose role in programmed cell death has been described since the nineties [34]. In addition, clinical studies suggested that anticoagulant treatments not only reduce cancer-related thromboembolic complications but may also interfere with both cancer development and progression [35].

In the light of this evidence, it is not surprising that we found an increase in pro-thrombin derived peptides in the serum of patients affected by colon cancer, in comparison with the healthy donors, even with reference to non-metastatic patients.

The complement system is involved in the innate immunity and has an important function in the surveillance against tumours [36], because it can be activated following the exposure to tumour antigens [37]. During one of the last steps of the complement cascade, factors I and H cleave C3b into iC3b generating the 17 amino acids peptide C3f [38]. Notwithstanding, complement activation has been considered for years a powerful tool for the elimination of cancer cells, a tumour-promoting role has recently been proposed for this part of the immune system [39]. Indeed, complement activation has been demonstrated to play a pro-tumorigenic role in inflammation-related colorectal cancer [40].

This observation is confirmed in the study carried out by Piao et al. where the authors describe the role of the fraction C5a of the complement system on metastasis spreading by means of macrophages stimulation [41]. This new point of view has convinced some researchers to hypothesize the concept of blocking complement for the treatment of cancer [42]. Consistently with the study of Wang et al. [10], we have identified the full length form of C3f fragment of the complement system (m/z 2021.08) as a candidate biomarker for CRC. However, several authors have reported an increase of the C3f levels in the biological fluids of patients affected by different cancer types [6,15,43,44].

Surprisingly, Bedin and co-workers described a lower intensity of the full length form of C3f fragment in serum of patients affected by early and late CRC in comparison to the control group [45]. However, this effect could be explained by the action of different N or/and C-terminal exopeptidases present in the serum samples. Indeed, a characteristic degradation pattern has been described for many of the proteins present in serum, including fibrinogen chains and complement components [13]. A confirmation of this hypothesis comes from a previous study of the former authors [46] in which they described the comparison of the plasma peptidome profile between healthy controls and patients affected by familial adenomatous polyposis (FAP). They found a clear increase of the full-length form of the C3f fragment of the complement system in FAP patients when compared with the healthy controls. However, they also observed a different pattern in the spectra obtained from the analysis of plasma samples from patients affected by colorectal cancer (CRC). In particular, they found a significant increase in signal m/z 1211.65 in CRC patients that is a degradation product of C3f. This observation leads us to suppose that the higher concentration of the C3f full-length form in patients with FAP compared to CRC is likely to be the result of a different peptidase activity, rather than a different complement system activation.

## 5. Conclusions

Though preliminary and obtained through a retrospective case-control study, our findings showed that MALDI/TOF MS peptidomic analysis of serum may provide a novel approach to CRC characterization. We observed that, while peptide C3f appears significantly increased only in serum from metastatic CRC patients, the increase of the prothrombin fragment is already evident in the serum of non-metastatic patients, suggesting its involvement in the early events of tumour progression.

## Figures and Tables

**Figure 1 diagnostics-10-01077-f001:**
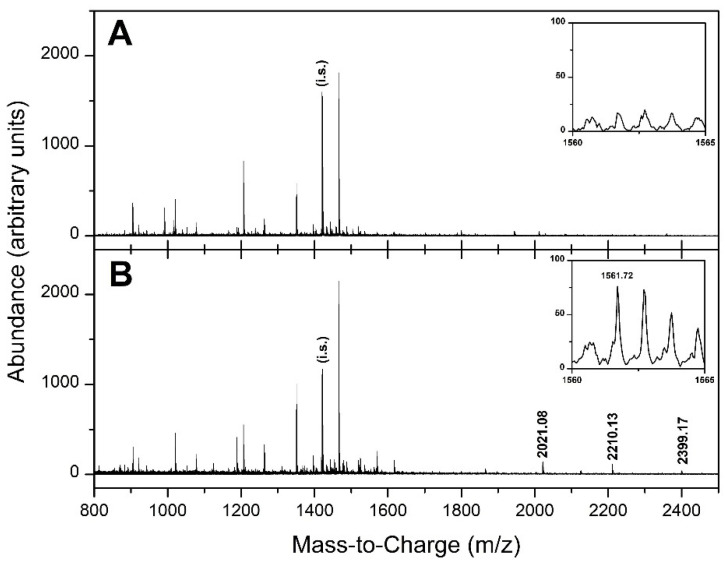
Serum sample spectra. The figure includes two spectra: healthy control (**A**) and metastatic colorectal cancer (CRC) patient (**B**). The intervals around the fragment at m/z 1561.725 have been zoomed (i.s.: internal standard).

**Figure 2 diagnostics-10-01077-f002:**
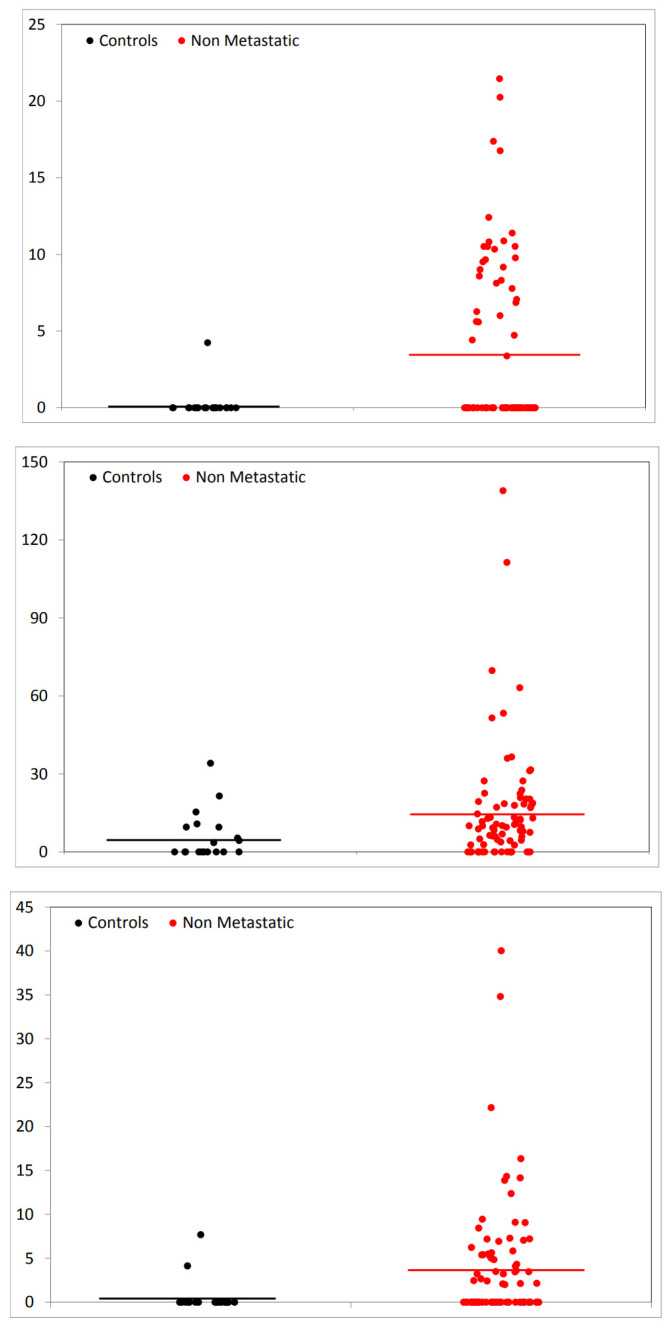
Dot plots representing abundance values for controls and non-metastatic patients. From top to bottom, plots refer to signals m/z 1561.72, 2021.08 and 2399.17. Dots on the left (black) represent controls, dots on the right (red) non-metastatic patients. The horizontal lines show the average values. Dots are randomly spread along the abscise axis for visibility. Scales are adapted to values.

**Figure 3 diagnostics-10-01077-f003:**
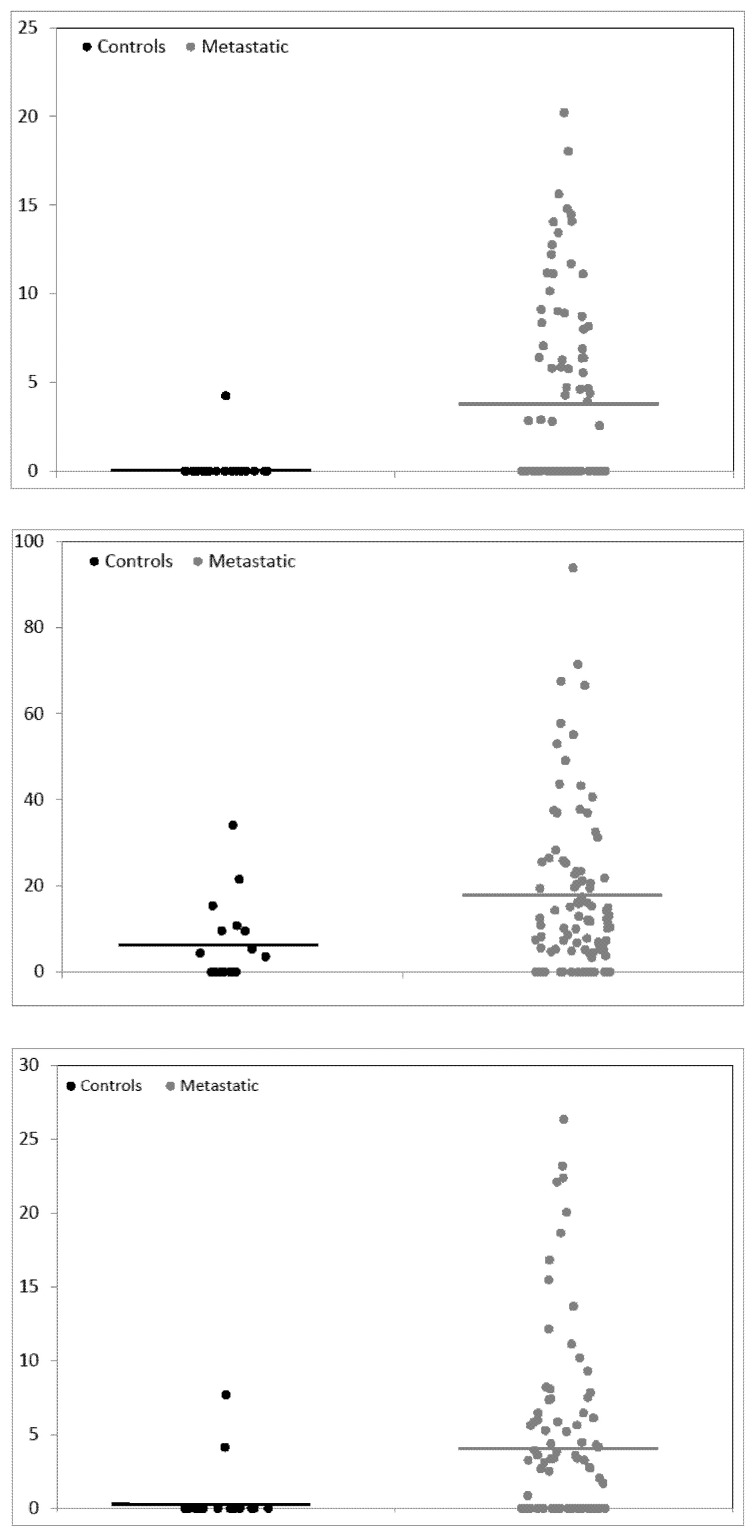
Dot plots representing abundance values for controls and metastatic patients. From top to bottom, plots refer to signals m/z 1561.72, 2021.08 and 2399.17. Dots on the left (black) represent controls, dots on the right (red) metastatic patients. The horizontal lines show the average values. Dots are randomly spread along the abscise axis for visibility. Scales are adapted to values.

**Table 1 diagnostics-10-01077-t001:** Characteristics of the groups of patients taking part in the study.

Type	Sample Size	Gender	Mean Age	Age Range
Metastatic CRC	92	28F/64M	71	34–93
Non-Metastatic CRC	80	26F/54M	71	44–89
Healthy controls	19	10F/9M	54	40–67

CRC: colorectal cancer, F: female, M: male.

**Table 2 diagnostics-10-01077-t002:** Results of significant analysis of microarrays (SAM) analysis on the original dataset.

***Controls vs. Non Metastatic***					
***Signal***	***Controls (n = 19)*** ***Mean (SE)***	***Cases (n = 80)*** ***Mean (SE)***	***Score***	***Fold Change***	***q-Value***
1561.720	0.22 (0.22)	3.67 (0.61)	2.371	16.408	*0.04490
2021.084	6.01 (2.12)	15.80 (2.54)	1.772	2.628	0.10264
2399.165	0.62 (0.45)	4.17 (0.79)	1.938	6.705	0.10264
***Controls vs. Metastatic***					
***Signal***	***Controls (n = 19)*** ***Mean (SE)***	***Cases (n = 92)*** ***Mean (SE)***	***Score***	***Fold Change***	***q-Value***
1561.720	0.22 (0.22)	3.97 (0.55)	2.634	17.786	* 0.03754
2021.084	6.01 (2.12)	17.52 (1.95)	2.489	2.915	* 0.03754
2399.165	0.62 (0.45)	4.28 (0.64)	2.246	6.877	* 0.03754

In the table, only those signals that were found significant in at least one of the analyses are reported. *q*-values are highlighted by an asterisk * when significant. SE: standard error, n: number of cases.

**Table 3 diagnostics-10-01077-t003:** Results of the bootstrap analysis.

*Controls vs. Non Metastatic*			*Controls vs. Metastatic*		
*Signal*	*q-Value*	*Bootstrap*	*Signal*	*q-Value*	*Bootstrap*
1561.720	* 0.04490	70/100	1561.720	* 0.03754	95/100
2021.084	0.10264	35/100	2021.084	* 0.03754	95/100
2399.165	0.10264	46/100	2399.165	* 0.03754	85/100

In this table, the results of the bootstrap analysis are reported in distinct sub-tables. Left: controls vs. non-metastatic patients; right: controls vs. metastatic patients. In each table, the leftmost value is the molecular weight of the signal, the central value is the *q*-value of the SAM analysis on the original data set, and the rightmost value is the number of occurrences of the signal find as significant (*p* < 0.05) in the SAM analyses on the new datasets. * denote the significant *q*-values (<0.05).

**Table 4 diagnostics-10-01077-t004:** Results from applying the multiple regression model to the peptides selected by the SAM analysis, adjusting by age and gender.

***Adjusted by Age***						
	***Non Metastatic vs. Controls***			***Metastatic vs. Controls***		
***Signal***	***Mean*** ***Difference***	***95% Confidence Interval***	***p-Value***	***Mean*** ***Difference***	***95% Confidence Interval***	***p-Value***
1561.720	2.58	0.13–5.02	0.039	4.85	2.68–7.03	<0.001
2021.084	Untested because not significant by SAM			9.51	1.12–17.91	0.026
2399.165	Untested because not significant by SAM			3.54	0.65–6.43	0.016
***Adjusted by Gender***						
	***Non Metastatic vs. Controls***			***Metastatic vs. Controls***		
***Signal***	***Mean*** ***Difference***	***95% Confidence Interval***	***p-Value***	***Mean*** ***Difference***	***95% Confidence Interval***	***p-Value***
1561.720	3.40	2.04–4.77	<0.001	3.75	2.55–4.95	<0.001
2021.084	Untested because not significant by SAM			10.84	5.57–16.11	<0.001
2399.165	Untested because not significant by SAM			3.33	1.86–4.81	<0.001

Signals at m/z 2021.084 and m/z 2399.165 were not tested for non-metastatic vs. controls comparison, because they were not found significant by SAM analysis.

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
