# Peer review of "The Activation of Prothrombin Seems to Play an Earlier Role than the Complement System in the Progression of Colorectal Cancer: A Mass Spectrometry Evaluation"

_diagnostics, 2020, doi:10.3390/diagnostics10121077_

Round 1

Reviewer 1 Report

The authors use peptidomic analysis to map differences in clotting factors across healthy patients, non-metastatic and metastatic CRC patients. They identify C3f elevation as a biomarker of metastatic CRC. CRC remains a global public health challenge, with significant mortality rates; therefore, the development of new biomarkers for early detection and patient stratification remains an attractive strategy.

I have a few comments below about the overall work;

  • A reference to the sample collection procedure and preparation for analysis would be good for future reference. This would be helpful for interlaboratory harmonization efforts.
  • The axis labels including units, and numbers for figure 2 are too small to read- these need to be enlarged. Also include a legend in the plot to ease differentiation between metastatic/non-metastatic sample sets.
  • There is not text between Figures 2 and 3. Same comments as for Figure 2 hold for Figure 3.
  • Are these findings consistent with any previous clinical reports or findings? If so, please refer to these. Have any preclinical models examined clotting factor role in tumour metastatic behaviour?
  • How specific is the biomarker to CRC vs other GI cancers or pro-inflammatory conditions? If this is non-specific, how will C3f be validated as a biomarker for CRC?

Author Response

Please note that the original review is in black and our replies are in red.

The authors use peptidomic analysis to map differences in clotting factors across healthy patients, non-metastatic and metastatic CRC patients. They identify C3f elevation as a biomarker of metastatic CRC. CRC remains a global public health challenge, with significant mortality rates; therefore, the development of new biomarkers for early detection and patient stratification remains an attractive strategy.

I have a few comments below about the overall work;

A reference to the sample collection procedure and preparation for analysis would be good for future reference. This would be helpful for interlaboratory harmonization efforts.

According with this request, after the following sentence:

“The CRC serum samples from patients were prepared from blood specimens collected before the start of any anticancer treatment”

we have added the following text:

"The samples were collected in the Basque Biobank for Research following a strict and standardized protocol to ensure the appropriate processing and storage that preserves a correct protein profile. Samples were obtained from peripheral blood. Tubes were centrifuged at 2500 rpm for 20 minutes at room temperature and placed in 500 ul aliquots.Note that the solid phase extraction of peptides has been described in details in the Materials and Methods section."

The axis labels including units, and numbers for figure 2 are too small to read- these need to be enlarged. Also include a legend in the plot to ease differentiation between metastatic/non-metastatic sample sets.

We have enlarged the units and related numbers and added a legend for this figure. Moreover, the layout of the figure has been changed. The three sections of the figure were laid in one horizontal row, which limited the available size for each section. Now they are arranged vertically so that each section is larger and more readable. The figure fits within a page, anyway.

There is not text between Figures 2 and 3. Same comments as for Figure 2 hold for Figure 3.

Figure 3 has been revised and improved as Figure 2. In order to improve the overall readability of the text and the legends, we have separated the figures, which are now in separate pages, by one short text paragraph.

Are these findings consistent with any previous clinical reports or findings? If so, please refer to these. Have any preclinical models examined clotting factor role in tumour metastatic behaviour?

We think we have provided, in the Discussion section, good evidence of the consistency of our data with the previous literature. In particular, a series of references that we believe are adequate for both C3f [6, 10, 33-38] and for prothrombin [18-26, 28] have been reported. Adams and co-workers [20] described the role of prothrombin in progression of colonic adenocarcinoma and in the formation of distant metastases but, interestingly, other authors [21] “established that the role of thrombin in cancer pathogenesis is not limited to late events, such as metastasis. Thrombin can also strongly influence the development of pre-cancerous lesions and subsequent tumour formation in the context of inflammation-driven colon cancer” (see Discussion section) confirming its early involvement in carcinogenic processes.

How specific is the biomarker to CRC vs other GI cancers or pro-inflammatory conditions? If this is non-specific, how will C3f be validated as a biomarker for CRC?

As stated, both in the Cover Letter and in the Manuscript, although both C3f (complement system) and prothrombin (coagulation cascade activation) have been associated with different types of cancer (see references), to our knowledge there are no studies demonstrating the quantitative incidence of these molecules during the progression of the CRC. The main goal of this study wasn’t the identification of a specific biomarker for CRC but, rather, the search for differences, at the serum level, between CRC metastatic and non-metastatic patients that could help clinicians in the characterization of patients with colorectal cancer.

Reviewer 2 Report

In the manuscript »The activation of prothrombin seems to play an earlier role than the complement system in the progression of colorectal cancer: a mass spectrometry evaluation« the authors report on their findings of serum analysis in metastatic and non-metastatic colorectal (CRC) patients vs. healthy controls. Their work is interesting and in accordance with the latest development of cancer diagnostic research. It is closely resembling the concept of liquid biopsy.

The authors nicely describe the rationale of their work in the introduction section of the manuscript. The number of the patients and controls population is appropriate, however they have not further specified, how they defined the metastatic (M1?; metastasis to the liver, lung, peritoneal carcinomatosis…?) and non-metastatic (N+ or only N0) patients. Apart from this minor shortcoming which doesn’t substantially reduce the core issue of the analysis the methods section in very well presented. The discussion section is clear and conclusions are compatible with the current concept of cancer progression. The reference section is relevant and sources reliable.

Author Response

Please note that the original review is in black and our replies are in red.

In the manuscript »The activation of prothrombin seems to play an earlier role than the complement system in the progression of colorectal cancer: a mass spectrometry evaluation« the authors report on their findings of serum analysis in metastatic and non-metastatic colorectal (CRC) patients vs. healthy controls. Their work is interesting and in accordance with the latest development of cancer diagnostic research. It is closely resembling the concept of liquid biopsy.

The authors nicely describe the rationale of their work in the introduction section of the manuscript. The number of the patients and controls population is appropriate, however they have not further specified, how they defined the metastatic (M1?; metastasis to the liver, lung, peritoneal carcinomatosis...?) and non-metastatic (N+ or only N0) patients.

According with this request, we have included in the manuscript the following sentence: “The assessment of metastatic and non-metastatic patients is based on the “Protocol for the Examination of Resection Specimens From Patients With Primary Carcinoma of the Colon and Rectum” of the College of American Pathologists (2020)”. The related bibliographic reference [Arch Pathol Lab Med. 2009 Oct;133(10):1539-51. doi: 10.1043/1543-2165-133.10.1539] was also added as [42]. For your convenience, please find here below the classification of each group, which has not been included in the manuscript.

Presence of distant Metastasis (pM)

pM1: Metastasis to one or more distant sites or organs or peritoneal metastasis is identified.

pM1a: Metastasis to one site or organ is identified without peritoneal metastasis.

pM1b: Metastasis to two or more sites or organs is identified without peritoneal metastasis.

pM1c: Metastasis to the peritoneal surface is identified alone or with other site or organ metastases.

Presence of absence of affected lymph nodes (N0 vs N1):

pNX: Regional lymph nodes cannot be assessed.

pN0: No regional lymph node metastasis.

pN1: One to three regional lymph nodes are positive (tumor in lymph nodes measuring ≥0.2 mm), or any number of tumor deposits are present and all identifiable lymph nodes are negative.

pN1a: One regional lymph node is positive.

pN1b: Two or three regional lymph nodes are positive.

pN1c: No regional lymph nodes are positive, but there are tumor deposits in the subserosa, mesentery, or nonperitonealized pericolic, or perirectal/mesorectal tissues.

pN2: Four or more regional lymph nodes are positive.

pN2a: Four to six regional lymph nodes are positive.

pN2b: Seven or more regional lymph nodes are positive.

Apart from this minor shortcoming which doesn't substantially reduce the core issue of the analysis the methods section in very well presented. The discussion section is clear and conclusions are compatible with the current concept of cancer progression. The reference section is relevant and sources reliable.